

# Expansion of great cormorant colony immediately increased isotopic enrichment in small mammals

Linas Balčiauskas[1], Raminta Skipitytė[1,2], Marius Jasiulionis[1], Laima Balčiauskienė[1], Vidmantas
Remeikis[2]

[1]Nature Research Centre, Akademijos 2, LT-08412 Vilnius, Lithuania
[2]Centre for Physical Sciences and Technology, Savanorių 231, LT-02300 Vilnius, Lithuania

*Correspondence to:* L. Balčiauskas (linasbal@ekoi.lt)



**Abstract.** Colonies of great cormorants (*Phalacrocorax carbo*) impact terrestrial ecosystems transporting nutrients from aquatic to terrestrial ecosystems. Deposited guano overload ecosystem with N and P, change soil pH, damage vegetation. However, ways how small mammals are impacted, are not known in details. We aimed to evaluate the effects of an expanding great cormorant colony, testing if the expansion immediately increases input of biogens to the forest ecosystem, and if influence of the colony is reflected in the basal resources (plants and invertebrates) and the hair of small mammals. Stable carbon and nitrogen isotope signatures were analysed in granivorous yellow-necked mice (*Apodemus flavicollis*), omnivorous bank voles (*Myodes glareolus*) and basal resources of animal and plant origin from the territory of a colony of great cormorants, situated near Baltic Sea, in West Lithuania. We found that biogens transferred by great cormorants to terrestrial ecosystems affected the potential foods of the small mammals and led to highly elevated and variable $\delta^{15}$N values. Increase of the colony in 2015 caused isotopic enrichment of the small mammals in the zone of expansion compared to 2014 levels. For most of the resources tested, the isotopic signatures in the established colony area were significantly higher, than in the ecotone (between colony and surrounding forest) and expansion zone with the first year on cormorant nests present. Surprisingly, in the colony $\delta^{15}$N values in plants (16.9 ± 1.1 ‰) were higher than in invertebrates (13.6 ± 0.4 ‰). In the ecotone $\delta^{15}$N values were 12.0 ± 1.4 ‰ in plants and 14.7 ± 0.04 ‰ in invertebrates, while in the expansion zone − 7.2 ± 3.0 ‰ and 9.9 ± 3.8 ‰, respectively. $\delta^{15}$N-rich resources led to increased $\delta^{15}$N values in the hair of *A. flavicollis* and *M. glareolus*. Thus, biogens from the great cormorant colony immediately affected small mammals through their food sources.

**Keywords**: $\delta^{13}$C and $\delta^{15}$N, *Apodemus flavicollis*, *Myodes glareolus*, *Phalacrocorax carbo*, colony increase, Lithuania

**Copyright statement**



# 1 Introduction

Impact of great cormorants (*Phalacrocorax carbo*) on terrestrial ecosystems is one of the biggest among birds breeding in colonies and transporting nutrients from aquatic to terrestrial ecosystems (Klimaszyk et al., 2015). Cormorant excreta change soil pH, N and P levels and damage vegetation (Kameda et al., 2006; Klimaszyk and Rzymski, 2016), decreasing diversity of

plants (Boutin et al., 2011), and affecting seed germination (Žółkoś and Meissner, 2008). Cormorant faeces may cover up to 80% of vegetation, with as many as 70 % of plant species disappearing in the established colonies, the rest being dominated by nitrophilous plants, such as elder (*Sambucus nigra*), common nettle (*Urtica dioica*), woodland groundsel (*Senecio sylvaticus*) and greater celandine (*Chelidonium majus*) (Goc et al., 2005; Klimaszyk et al., 2015). The abundance and diversity of herbivorous insects also decreases, while other arthropod groups may be abundant (Kolb et al., 2012).

With regard to great cormorant colonies and small mammals investigations are scarce. In previous studies, we have discovered that great cormorant colony in Juodkrantė, Lithuania, change the ecology of the small mammals inhabiting the territory, decreasing their diversity and abundance (Balčiauskienė et al., 2014), changing population structure and fitness (Balčiauskas et al., 2015) – all changes indicating poor habitat, and heightening values of $\delta^{13}$C and $\delta^{15}$N in their hair (Balčiauskas et al., 2016). Our continuing interest in the great cormorant colony in Juodkrantė was maintained by the fact that it's area and number

of breeding pairs increased in 2015 due to the absence of deterrent measures. Cormorants built nests in the formerly uninhabited territory, giving opportunity to evaluate immediate effect of the colony formation. As number of nests increased in the colony, we hypothesized that stable isotope values will also increase in small mammal hair.

Measures of limiting breeding success in Juodkrantė great cormorant colony were started in 2004 (Knyva, unpublished) and they withhold colony from expansion. In 2015 measures were not applied, resulting colony growth. First nests appeared in the

area, which was free of cormorants in 2014, thus, was used as control zone in Balčiauskas et al., (2016). In 2015 we repeatedly examined $\delta^{13}$C and $\delta^{15}$N distribution in a great cormorant colony, this time including in plants and invertebrates as expected diet sources of small mammals. We analysed the isotope composition in small mammal hair during a period of cormorant colony growth, comparing isotopic signatures in small mammal hair between 2014 and 2015. We do not analyse species, gender and age based differences of stable isotope values in the hair of small mammals. The aim was to evaluate the effects of

the transfer of biogens from the aquatic to terrestrial ecosystem by an expanding great cormorant colony. We tested (i) if the expansion of the great cormorant colony immediately increases input of biogens to the forest ecosystem, and (ii) if influence of the great cormorant colony is reflected in the basal resources (plants and invertebrates) and the hair of small mammals. Novelty of our investigation was in evaluating speed of the impact of great cormorant colony on small mammals. The results give new insight to understanding how fast biogenic pollution is transferred and what the consequences are to small mammal

ecology.



# 2 Material and methods

## 2.1 Study site

In 2015, small mammals were trapped and samples of their possible foods were collected in the territory of the biggest colony of great cormorants (*Phalacrocorax carbo*) in Lithuania (Fig. 1), situated near Juodkrantė settlement (WGS 55° 31' 14.22" N,

21° 6' 37.74" E). The colony is in Kuršių Nerija National Park, which has been a UNESCO World Heritage site since 2000. Colony existed in 19 century, but due to persecution disappeared in 1887 and returned only after 100 years. Breeding of cormorants was again registered in 1989, and in 1999 there were over 1000 breeding pairs. In 2004 number of breeding pairs reached 2800. Over 3500 nests have been recorded in the colony each year since 2010, with the exception of 2014 when, due to control measures (firing petards in the nesting period), the number of successful pairs was under 2000. In 2015, control

measures were not used and the number of breeding pairs reached 3800.

Three zones were defined for this study, colony, ecotone and expansion zone. Colony included area with the highest concentration of nests and the area of former active influence with dead trees. Zone of ecotone was situated between colony and forest to the south (Fig 1a). It's position did not change in 2011–2014. In 2015 very few nests appeared in the ecotone. However, in 2015 the area of the colony expanded about 3 ha northward (Fig 1b). Nests and droppings appeared in the

expansion zone, however, trees had no visible influence of the birds. In 2013–2014 expansion zone was used as trapping control (Balčiauskas et al., 2015; 2016), thus, we compared results to find colony influence after the first year of cormorant breeding in the formerly free territory.

## 2.2 Small mammal sampling

Small mammals were trapped in September 2015, using lines of 25 snap traps placed every 5 m. Expansion and ecotone zones

had two such lines each, and six lines were located in the great cormorant colony (Fig. 1a). Baited with bread and sunflower oil, the traps were open for three days and checked every morning. Trapping effort was equal to 750 trap days and 125 individuals were trapped. Study was conducted in accordance with the principles of Lithuanian legislation for animal welfare and wildlife.

The dominant species were yellow-necked mice (*Apodemus flavicollis*) and bank voles (*Myodes glareolus*), while harvest mice

(*Micromys minutus*), root voles (*Microtus oeconomus*) and short-tailed voles (*M. agrestis*) were trapped in very low numbers. Two shrew species, common (*Sorex araneus*) and pygmy shrews (*S. minutus*), were trapped occasionally. No small mammals were trapped in areas of the great cormorant colony that contained the highest concentration of nests and had experienced long term influence. In the expanding part of the colony the only species, *A. flavicollis* was trapped in numbers.

Trapped rodent species differ in food preferences, ranging from herbivory in *Microtus* to granivory in *Apodemus*, *Micromys*

and omnivory in *Myodes* (Butet and Delettre, 2011; Čepelka et al., 2014). In general, small mammals are mostly omnivorous (Nakagawa et al., 2007), though *Sorex* typical consume invertebrates (Makarov and Ivanter, 2016).



Individuals were measured, sex and age were recorded during dissection as described elsewhere (Balčiauskas et al., 2015). The age and sex composition of the dominant species is presented in Table 1.

## 2.3 Baseline sampling

Isotopic signatures were evaluated and isotopic baselines were established from possible dietary items. In 2015 we collected samples of the possible food items at the locations where the small mammals were trapped or the closest available place. In the most affected zones plant diversity was extremely restricted, with a few nitrofilous species present. In total, 45 plant and 9 invertebrate samples were collected. Five litter samples and seven samples of great cormorant feathers and eggshells were also collected.

Plant samples included leaves of greater celandine (*Chelidonium majus*), sedges (*Carex* sp.), raspberry (*Rubus idaeus*), rush (*Juncus* sp.), blackberry (*Rubus fructicosus*) and billberry (*Vaccinium myrtillus*), leaves and berries of elder (*Sambucus nigra*), alder buckthorn (*Rhamnus frangula*), European barberry (*Berberis vulgaris*), and oak (*Quercus robur*) acorns. Invertebrate samples included coprophagous dung beetle (*Geotrupes stercorarius*), herbivorous dark bush-cricket (*Pholidoptera griseoaptera*), predatory ground beetle (*Carabus* sp.) and omnivorous land slug (*Deroceras* sp.). Quantifying of the different foods by volume was not done. Unfortunately, we have no data on isotopic signatures in basal resoources from the pre-expansion period in 2014.

## 2.4 Stable isotope analysis

We used hair of the rodents as metabolically inert samples, preserving the isotopic record of the animal's diet (Crawford et al., 2008; Bauduin et al., 2013). Hair samples were cut off with scissors from between the shoulders of all trapped specimens of *A. flavicollis* and *M. glareolus* (88 and 29 individuals, see Table 1). Each sample was placed separately in bags and stored dry. Samples were weighed with a microbalance and packed in tin capsules. Few individuals of these species survive longer than 1 year; over-winter mainly autumn-born ones (Bobek, 1969), thus, our samples represent cormorant influence of the year when rodents were trapped.

Environmental samples (including plants, litter, invertebrates and great cormorant feathers and eggs) were stored in a refrigerator at below -20°C prior to preparation and analysis. Samples were dried in an oven at 60°C to a constant weight for 24–48 hours and then homogenized to a fine powder (using mortar and pestle and a Retsch mixer mill MM 400). Pretreatment of hair and other samples was not used, as after testing it gave no change of results. Feathers were cleaned with acetone and deionized water prior to measurements. Feather samples were clipped from the vane avoiding the rachis.

Stable isotope ratios ($\delta^{13}$C and $\delta^{15}$N) were measured using an elemental analyzer (EA) coupled to an IRMS (Flash EA1112; Thermo Delta V Advantage, Thermo Scientific, USA). Stable isotope data are reported as $\delta$ values, according to the formula $\delta$X = [R$_{sample}$/R$_{standard}$ - 1] × 10$^3$, where R$_{sample}$ = $^{13}$C/$^{12}$C or $^{15}$N/$^{14}$N of the sample, R$_{standard}$ = $^{13}$C/$^{12}$C or $^{15}$N/$^{14}$N of the standard.



5 % of samples were run in duplicate. The equipment parameters and measurement quality control are detailed elsewhere (Balčiauskas et al., 2016).

## 2.5 Statistical analysis

Normality of distribution of $\delta^{13}$C and $\delta^{15}$N values was tested using Kolmogorov-Smirnov's D. As not all values of $\delta^{13}$C and $\delta^{15}$N were distributed normally, the influences of species and the zone of the colony on the carbon and nitrogen stable isotope values in the mammal hair were tested using nonparametric Kruskal-Wallis ANOVA. Independent groups were compared with the same Kruskal-Wallis multiple comparisons procedure (Electronic, 2017). Differences in $\delta^{13}$C and $\delta^{15}$N ratios between 2014 (data from Balčiauskas et al., 2016) and 2015 were tested by multivariate Hotteling's $T^2$ test. The minimum significance level was set at $p < 0.05$. Calculations were performed using Statistica for Windows, ver. 6.0.

Environmental samples were analyzed by object group (cormorant, litter, invertebrates, plants) and by the zone (expansion, ecotone, colony). Isotopic baselines were calculated using basal resources as possible foods for rodents grouped according to their origin. Reported values are arithmetic means with SE of the $\delta^{15}$N and $\delta^{13}$C for all basal resources mentioned above.

# 3 Results

## 3.1 $\delta^{13}$C and $\delta^{15}$N values in the hair of small mammals inhabiting great cormorant colony

Distribution of both $\delta^{13}$C and $\delta^{15}$N values in the hair *A. flavicollis* was not normal, while in *M. glareolus* distribution of $\delta^{15}$N values was not normal, but distribution $\delta^{13}$C values corresponded to normal (Fig. S1). Outliers from the normal distribution were values, registered in the expansion zone. Kruskal-Wallis ANOVA demonstrated that the distribution of stable isotope values was influenced not only by zone, but also by species of small mammal. These factors together significantly influenced the distribution of $\delta^{15}$N ($r^2 = 0.31$) and $\delta^{13}$C ($r^2 = 0.26$, F both $p < 0.0001$).

In 2015, influence of the zone (both species pooled) was significant for the distribution of $\delta^{15}$N (Kruskal-Wallis ANOVA, $H_{2,119} = 18.62$, $p = 0.0001$) and $\delta^{13}$C ($H_{2,119} = 6.30$, $p = 0.043$). Between-species differences of the stable isotope values in the hair of rodents in the colony area were highly significant for $\delta^{13}$C ($H_{1,119} = 21.69$, $p < 0.0001$) and for $\delta^{15}$N ($H_{1,119} = 6.67$, $p = 0.01$). $\delta^{15}$N values were highest in hair of *M. glareolus* trapped in the ecotone and colony zones, while in *A. flavicollis* – in the expansion zone. $\delta^{13}$C signatures in the hair of *A. flavicollis* were higher than in *M. glareolus* in all territories, including the expansion zone (Table 2).

With the development of the great cormorant colony in 2015, the isotopic signatures, mostly $\delta^{15}$N, in dominant small mammal hair grew compared to 2014, though not all differences are significant (Table 2).



## 3.2 Basal resources

Average baseline data of plants and invertebrates of the expansion zone, ecotone and colony (Table 3) showed the highest differences in $\delta^{15}$N of plants (Kruskal-Wallis ANOVA, $H_{2,45}$ = 13.89, p = 0.001) but not invertebrates ($H_{2,9}$ = 2.76, p = 0.25). In plants, $\delta^{15}$N values were highest in the colony (difference from expansion zone, p < 0.002; difference from ecotone, p =

0.062). $\delta^{13}$C values showed no significant differences between zones in either plants or in the invertebrates (Table 3).

Out of ten plant species, the most $^{15}$N-enriched were: *Chelidonium majus*, $\delta^{15}$N = 19.6 ±2.0 ‰ (from 8.4 ‰ in the expansion zone to 25.7 ‰ in the colony), *Sambucus nigra*, $\delta^{15}$N = 16.9 ±2.6 ‰ (from 9.2 ‰ in the ecotone to 22.5 ‰ in the colony), *Juncus* sp., $\delta^{15}$N = 13.8 ±4.7 ‰ and *Rhamnus frangula*, $\delta^{15}$N = 13.2 ±1.8 ‰ (from 8.1 ‰ in the expansion zone to 16.3 ‰ in the colony). $\delta^{15}$N values in *Carex* sp. (average $\delta^{15}$N = 12.3 ±0.9 ‰) did not differ significantly across zones.

The most $\delta^{13}$C-enriched plants were *Carex* sp. with $\delta^{13}$C = -26.7 ±0.7 ‰ and *Sambucus nigra*, $\delta^{13}$C = -27.5 ±0.6 ‰, values in the expansion zone and colony did not differ. *Chelidonium majus* was among the less $\delta^{13}$C-enriched plants, $\delta^{13}$C = -29.5 ±0.4 ‰.

Of the investigated invertebrates, the most $\delta^{15}$N enriched were *Carabus* sp. and slugs, $\delta^{15}$N = 14.9 ‰ and 14.0 ‰ respectively. Dung beetles showed the highest variation of $\delta^{15}$N, being two times lower in the expansion zone than in the colony (6.2 ‰

versus 13.3 ‰). The stable carbon isotope ratio in slugs was about 1 ‰ higher than in arthropods.

$\delta^{15}$N values in the litter were highest in the colony ($\delta^{15}$N = 16.3 ‰), followed by ecotone and expansion zones ($\delta^{15}$N = 8.2 ‰ and 7.3 ‰, respectively). $\delta^{15}$N values in the cormorant eggshells and feathers were lower than in rodent hair, and not depending from zone (Fig. S2).

## 3.3 Comparison of isotopic signatures in the hair of small mammals and in possible

## diet resources

Comparison of isotopic signatures in the hair of *A. flavicollis* and *M. glareolus* with baseline $\delta^{13}$C and $\delta^{15}$N values in possible food sources from the expansion, ecotone and colony zones showed that differences were related to bird influence.

In the expansion and ecotone zones, invertebrate isotopic signatures were higher than in plants in terms of both $\delta^{13}$C and $\delta^{15}$N. In great cormorant colony, most plants were highly enriched in $^{15}$N due to over-enrichment and tended to have $\delta^{15}$N values

well above the invertebrate $\delta^{15}$N range (Fig. 2).

Compared to average plant and invertebrate baseline values, higher average of $\delta^{15}$N and $\delta^{13}$C values in the hair of *A. flavicollis* and *M. glareolus* was related to the zone where rodents were trapped (Table 4). For *A. flavicollis* trapped in the expansion and ecotone zones, the average $\delta^{15}$N was over 5 ‰ higher than the plant baseline, though the plant baseline in the colony was higher than $\delta^{15}$N in the hair. Compared to the invertebrate baseline, in the rodent hair enrichment of $^{15}$N was 2.5–3 ‰.

Concerning *M. glareolus*, $^{15}$N enrichment was highest in the ecotone when compared to the plant baseline, but highest in the colony zone when compared to invertebrate baseline. As for $^{13}$C, enrichment was highest in the ecotone zone for both *A. flavicollis* and *M. glareolus* (Table 4).



# 4 Discussion

## 4.1 Immediacy of the great cormorant colony impact

Our main finding showed that great cormorants influenced small mammals in the very first year of the appearance of breeding colony, and possible food objects (plants and invertebrates) also were subjected to increased $\delta^{13}$C and $\delta^{15}$N concentrations. Moreover, in 2015 – the year of colony increase and expansion – we found, that increased influence of the great cormorant colony limited even small mammal distribution. Small mammals were not trapped in the area with the highest concentration of nests. However, representatives of three small mammal species were trapped in the same place in earlier years, when number of nests was lower due to scaring measures. In the expanding part of the colony single individual of *M. glareolus* was trapped in 2015, and the only species, *A. flavicollis* was trapped in numbers. In 2014, before colony expansion, four small mammal species were trapped in the same area.

## 4.2 Biogenic pollution is disclosed by stable isotope concentration in the hair of small mammals

Investigations of the influence of great cormorant colonies have recently received more attention (see Ishida, 1996; Goc et al., 2005; Kameda et al., 2006; Nakamura et al., 2010; Klimaszyk et al., 2015; Klimaszyk and Rzymski, 2016). However, the impact of such colonies on plant and animal species is insufficiently investigated (see Bostrom et al., 2012; Kolb et al., 2012). Small mammal ecology in the colonies was for the first time investigated in Lithuania (Balčiauskienė et al., 2014; Balčiauskas et al., 2015). Some aspects of isotopic enrichment of small mammals in great cormorant colonies were reviewed in Balčiauskas et al., (2016). We found, that biogenic pollution of the birds reach dominant species of small mammals. Stable stable isotope ratios in their hair depend on the power of cormorant influence, being strongest in the area with cormorant nests (see data in Table 2), and depending on the colony size. However, we had no data of stable isotopes in basal resources and we were not aware, how fast may be such influence.

## 4.3 Stable isotopes show small mammal diet differences in various zones of cormorant colony

Stable isotope analysis (SIA) can be used to investigate the trophic structure of food webs and various aspects of animal diet (Boecklen et al., 2011; Koike et al., 2016). SIA using mammal hair is a method suitable for diet analysis and for comparing intrapopulation groups as well as different species. When analyzing diet, hair isotopic signatures are compared with the signatures of possible food sources (Bauduin et al., 2013). In identifying diet sources, carbon and nitrogen isotope ratios are most widely used. The tissues of animals do differ in isotopic composition as a result of differences in their diet. Nitrogen values help in identifying trophic position, since these values increase by 3–5 ‰ between levels of the food chain (Kelly, 2000; Smiley et al., 2016).



The stable isotope ratio in the hair reflects the dietary isotope composition and trophic level, depending on the ingested food (Cassaing et al., 2007). As hair of the rodents trapped in the Juodkrantė colony, ecotone, and expansion zone differed in $\delta^{15}N$ and $\delta^{13}C$ values, rodents obviously consumed foods with different isotopic signatures, the more diverse diet being in the expansion zone as reflected by much higher variance of $\delta^{13}C$ values. Enrichment of plants and invertebrates was strongest in

the territory of colony (see Fig. 2).

Species-related differences in the isotopic signatures of the two dominant rodent species may be explained by diet differences and microhabitat use, both supporting coexistence (Stenseth et al., 2002; Cassaing et al., 2013). Previous experience with live-trapping and marking of *A. flavicollis* in the cormorant colony (Jasiulionis, unpubl.) let us conclude, that migration was very limited: we did not found marked animals using several zones in the same year. In the resource-scarce territory of the great

cormorant colony, any spatial segregation could lead to changes in the diet of rodents. Considering isotopic signatures in the hair of small mammals as dietary proxies (according Fernandes et al., 2014) reflecting proteins of the food sources (Perkins et al., 2014), we found that diets differ in the various zones of the colony and depending on the small mammal species. Therefore, differential exploitation of resources minimizes competition (Bauduin et al., 2013).

## 4.4 Pathways of small mammal enrichment in stable isotopes

Possibility of rodent enrichment in $\delta^{15}N$ from eating cormorant tissues is not clear. Dead chicks, broken eggs and eggshells are constantly present on the ground underneath the nests in the breeding season, so could serve as food source. Moreover, $\delta^{15}N$ values in great cormorant eggshells and feathers were lower than $\delta^{15}N$ values in the hair of rodents from the same zone (See Fig. S2). Furthermore, there are observations of *Microtus* or *Myodes* voles eating auklets' eggs and chicks (Drever et al., 2000). In our case, difference in $\delta^{15}N$ between cormorants and rodents is not big, questioning possibility of consumption of

cormorant in any significant amount. So we support the opinion that cormorant influence is mediated through disturbance of food resources (Millus and Stapp, 2008).

Two possible pathways of marine nitrogen are (1) through guano-fertilised plants or (2) invertebrates that have fed on guano, guano-fertilised plants or cormorant remains (Harper 2007). According to Szpak et al. (2012), $^{15}N$ enrichment of plants may range from 11.3 to 20 ‰ after fertilization by guano of seabirds. Plants enriched in guano $^{15}N$ may occur at distances exceeding

100 m from nesting sites and colonies (Millus and Stapp, 2008). Few plant samples (*Carex* and *Sambucus*) were highly enriched in $^{15}N$ at the expansion zone of Juodkrantė colony in 2015, first year of cormorant presence.

Rodents usually eat foods that are most abundant (Bauduin et al., 2013) or have preferences characteristic to the species (Fisher and Türke, 2016; Schneider et al., 2017), however, choices in the Juodkrantė great cormorant colony are limited to several plant species (mainly nitrophilous) and invertebrates. Nitrophilic plants usually grow abundantly, being the food source for

herbivores living in the territory of a colony (Cassaing et al., 2007). Even after birds cease to use territory, isotopic signatures of the litter and plants remain high (Kameda et al., 2006). Enrichment of plants by $^{15}N$ is a result of uptake of nitrogen from biopolluted soil enriched by marine-derived N from great cormorant excreta. In such situation, $\delta^{15}N$ is not a straightforward



indicator of the trophic level (Drever at al., 2000). On seabird islands, herbivores often exhibit heightened $\delta^{15}$N signatures (Stapp et al., 1999; Drever et al., 2000).

Typically, diet-tissue fractionation is from 2.5 ‰ to 3.4 ‰ for nitrogen (Perkins et al., 2014). Trophic fractionation from 3 to 5 ‰ for nitrogen occurs at every trophic level in seabird colonies (Cassaing et al., 2007). However, in Juodkrantė it may exceed 5 ‰ in comparison to plants. Trophic fractionation for carbon of *A. flavicollis* was in predictable level, up to 4.6 ‰ in comparison to plants, and up to 3 ‰, compared to the invertebrate baseline. Enrichment in $^{13}$C of *M. glareolus* was lower, up to 3.6 ‰ compared to plants, and up to 1.9 ‰, compared to the invertebrate baseline (see Table 4). These values are similar to or even higher than those observed by Sponheimer et al. (2003), with a mean diet–hair fractionation of +3.2 ‰ and a range of +2.7 to +3.5 ‰ in mammalian herbivores.

# 6 Conclusions

This study seek to understand, how influence of biologic pollution from the great cormorant colony reach small mammals, and how fast this influence is registered after birds appear in the territory. The general conclusion is that in Juodkrantė, Lithuania, the great cormorant colony affected the terrestrial ecosystem starting from autotrophs and ending with the consumers (two species of rodents). Increase of the number of breeding pairs in 2015 led to increased $\delta^{13}$C and $\delta^{15}$N values in the hair of *Apodemus flavicollis* in the territory of the colony, ecotone and in expansion zone. In the colony expansion zone influence was visible after the first year of nest appearance. In the resource-limited territory under the great cormorant nests, differences in isotopic signatures were related to species of rodents, pointing out to the differences in their diet. We conclude that despite the nutrient transport from water to land ecosystems by great cormorants, their influence is indirect, resulted by biological pollution from guano on rodent foods. Our results show that scaring cormorants from the colonies may have opposite effect. Negative impact of the emerging colony is spanning through entire ecosystem and reaching mammals in the first year.

# Data availability

Data used in this paper are available upon request from the corresponding author.

# Supplement link



## Author contributions

MJ, LB1 and LB2 trapped small mammals and collected baseline data. MJ and RS analysed stable isotopes. LB1 analysed data statistically. LB1 and LB2 wrote first draft. All authors provided substantial input to the design of the study and discussion of the results.

## Conflict of interest

The authors declare that they have no conflict of interest.

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





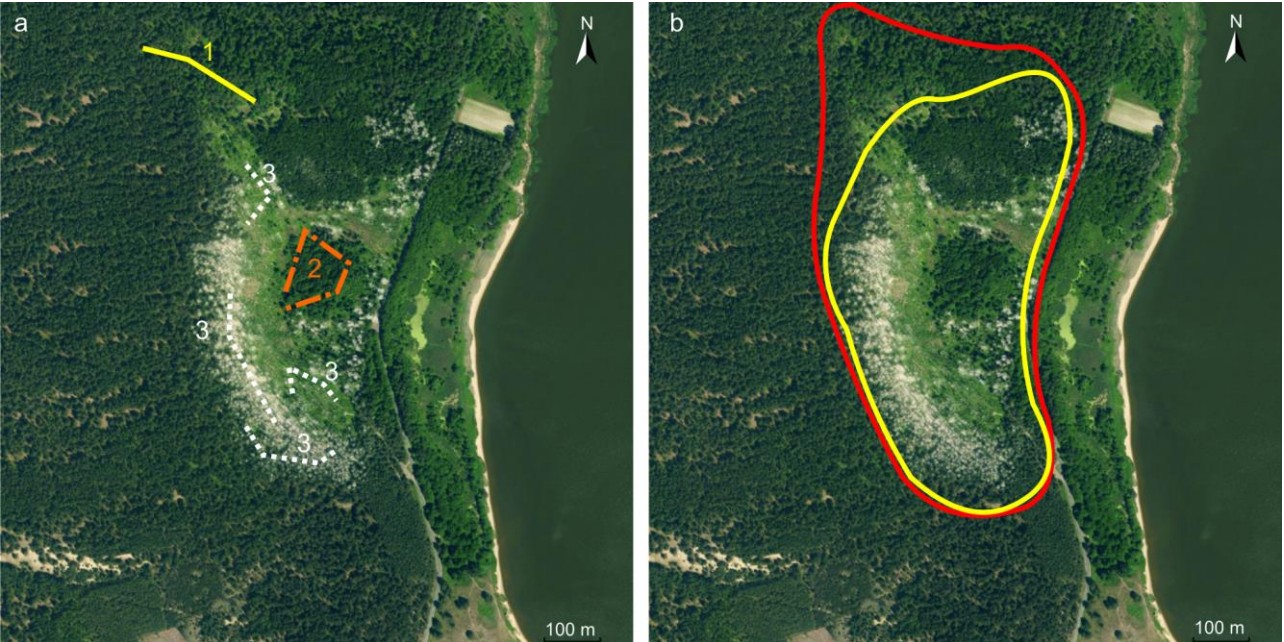

**Figure 1. Trapping design in the great cormorant colony in Juodkrantė (a) and colony expansion in 2015 (b). Zones: 1 – expansion, 2 – ecotone, and 3 – the colony in 2015 (a); yellow line – colony area in 2014, red line – colony expansion in 2015 (b).**



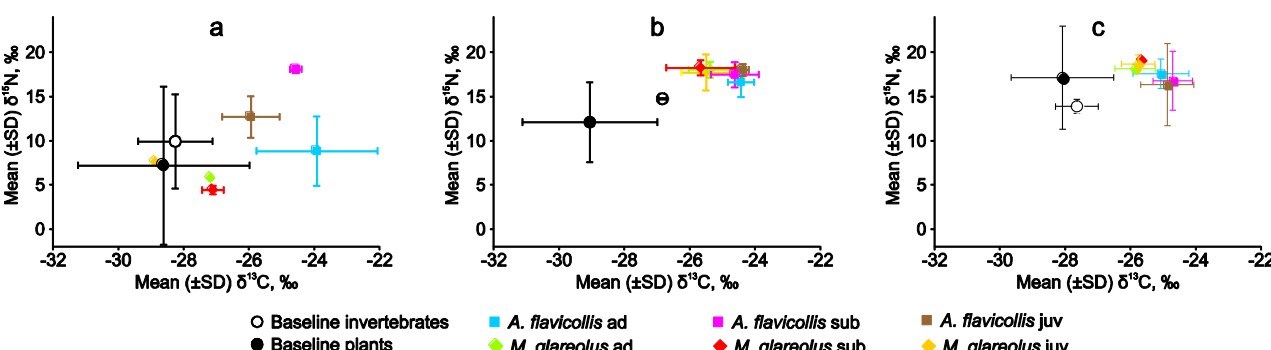

**Figure 2. Isotopic signatures of potential animal and plant foods compared with isotopic signatures in the hair of age groups of** *Apodemus flavicollis* **and** *Myodes glareolus***, trapped in the expansion (a) and ecotone (b) zones, and Juodkrantė great cormorant colony (c) in 2015. Ad – adults, sub – subadult animals, juv – juveniles.**



**Table 1 Age and sex structure of the two dominant small mammal species trapped in the Juodkrantė great cormorant colony in 2015.**

| Species | Sex | Adult | Subadult | Juveniles |
|---|---|---|---|---|
| *Apodemus flavicollis* | Male | 18 | 17 | 11 |
| | Female | 12 | 15 | 15 |
| *Myodes glareolus* | Male | 7 | 5 | 5 |
| | Female | 6 | – | 6 |

**Table 2 Central position (mean±SE, ‰) of stable isotope ratios in the hair of *Apodemus flavicollis* and *Myodes glareolus* trapped in the Juodkrantė colony of great cormorants. Data for 2014 recalculated from Balčiauskas et al. (2016). Significance of differences according Hotteling's $T^2$ multivariate test, * – $p < 0.1$, ** – $p < 0.05$, *** – $p < 0.001$.**

| Species | Zone | Year | $\delta^{13}C$, ‰ ± SE | $\delta^{15}N$, ‰ ± SE |
|---|---|---|---|---|
| *Apodemus flavicollis* | Control | 2014 | -24.20±0.08 | 12.26±1.04 |
| | Expansion | 2015 | -25.30±0.33* | 12.96±0.94 |
| | Ecotone | 2014 | -24.37±0.13 | 15.97±0.45 |
| | | 2015 | -24.51±0.10 | 17.16±0.26** |
| | Colony | 2014 | -24.08±0.12*** | 16.52±0.90 |
| | | 2015 | -24.82±0.12 | 16.67±0.50 |
| *Myodes glareolus* | Control | 2014 | -25.82 | 14.30 |
| | Expansion | 2015 | -28.02±0.85 | 6.70±0.91 |
| | Ecotone | 2014 | -24.85±0.30 | 17.48±0.64 |
| | | 2015 | -25.49±0.16 | 17.87±0.30 |
| | Colony | 2014 | -24.89±0.31* | 17.99±0.91 |
| | | 2015 | -25.72±0.17 | 18.17±0.26 |

**Table 3 Isotopic signatures of basal resources (plants and invertebrates) in the Juodkrantė great cormorant colony, ecotone and expansion zones in 2015**

| Basal resources | Zone | n | $\delta^{13}C$, ‰ ± SE | $\delta^{13}C$, ‰ min–max | $\delta^{15}N$, ‰ ± SE | $\delta^{15}N$, ‰ min–max |
|---|---|---|---|---|---|---|
| Plants | Expansion | 9 | -28.6 ±0.9 | -32. 8 – -25.2 | 7.2 ±3.0 | -3.3 – 25.1 |
| | Ecotone | 10 | -29.1 ±0.7 | -31.6 – -25.2 | 12.0 ±1.4 | 8.0 – 22.5 |
| | Colony | 26 | -28.1 ±0.3 | -31.4 – -25.2 | 16.9 ±1.1 | 8.4 – 27.7 |
| Invertebrates | Expansion | 2 | -28.3 ±0.8 | -29.1 – -27.5 | 9.9 ±3.8 | 6.2 – 13.7 |
| | Ecotone | 2 | -26.9 ±0.1 | -26.9 – -26.8 | 14.7 ±0.0 | 14.6 – 14.7 |
| | Colony | 5 | -27.6 ±0.3 | -28.4 – -26.5 | 13.6 ±0.4 | 12.8 – 14.9 |





**Table 4 Nitrogen ($\varDelta\delta^{15}$N) and carbon ($\varDelta\delta^{13}$C) trophic fractionation between *Apodemus flavicollis* and *Myodes glareolus* and their possible food sources in the Juodkrantė great cormorant colony.**

| Zone | Baseline for comparison | $\varDelta\delta^{15}$N, ‰ | | $\varDelta\delta^{13}$C, ‰ | |
|---|---|---|---|---|---|
| | | *A. flavicollis* | *M. glareolus* | *A. flavicollis* | *M. glareolus* |
| Expansion | Plants | 5.78 | -0.48 | 3.31 | 0.59 |
| | Invertebrates | 3.05 | -3.21 | 2.95 | 0.23 |
| Ecotone | Plants | 5.15 | 5.86 | 4.58 | 3.60 |
| | Invertebrates | 2.49 | 3.20 | 2.34 | 1.37 |
| Colony | Plants | -0.23 | 1.27 | 3.32 | 2.41 |
| | Invertebrates | 3.11 | 4.61 | 2.76 | 1.86 |