# Peer review of "Immediate increase in isotopic enrichment in small mammals following the expansion of a great cormorant colony"

_Biogeosciences, 2017_

## Referee Comment (RC1) · Anonymous Referee #1 · 9 Feb 2018

This study examined the effects of great cormorant's colonies on nutrient inputs into forest ecosystems using stable N and C isotopes. The goals of this study were to test (1) whether the expansion of the cormorant colony immediately affected the nutrient input to a forest ecosystem and (2) whether the nutrient derived from cormorants influenced the isotopic signatures of the basal food resources and small mammals. It is known that recent rapid growth of great cormorant colonies had negative impacts on many terrestrial ecosystems. In addition, the linkage between the aquatic and terrestrial ecosystems has been one of the main topics in ecological studies. Therefore, this study might attract many readers from the fields of both applied and basic sciences. However, I have some concerns about the design and the novelty of this study.

[Figure]

The first goal of this study as mentioned in Page 3 L 26 seems too obvious. It should be clear that the nutrient input increases as the number of great cormorants and the faecal deposit become greater in the new habitat. I suppose that the authors intended to mention that they aimed to show how rapidly nitrogen derived from the great cormorants were used by primary produces and the consumers at higher trophic levels. Even if this were one of their goals, the presented data would seem not enough to achieve the goal. The data used for this purpose were the isotopic signatures of two species of small mammals at a single control site in 2014 and at the same site but inhabited by the great cormorants in 2015. I think that replicated study sites would be necessary to examine statistically the effects of colony expansion on the isotopic signatures of consumers. In addition, the statistical results (Table 2) showed that there was almost no significant difference in 15N of the two small mammals between before-expansion (2014) and after-expansion (2015). It seems at least to me that the small mammals were not dependent on the cormorant-derived N, which is not consistent with the conclusion of this study (e.g., P2 L15). Please explain from which datasets the conclusion was drew. Regarding the second goal, the authors had already shown that the great cormorant colonies significantly affected the isotopic signatures of the small mammals in their previous study published in 2016. Therefore, although I recognize the importance of datasets of the basal food sources presented by this study, the goal and the obtained results seem highly confirmatory.

Minor comments: P2 L2: "and" damage P2 L12: It was not surprising to me. It simply indicates that the plant used more cormorant-derived N than the invertebrates. P3L14: It would be better to describe the background and the importance of this study based not on the authors' group interest, but on the scientific interest. P3L23: This sentence (gender and age) is not necessary here. P5L2: If the authors think this information (Table 1) is necessary, please provide it as a supplementary file. P8L3: Which datasets showed the influences of great cormorants on small mammals? Please explain. P8L24-P9L13: This subsection seems not directly related to the aims of this study. P9L20: This sentence is unclear. Please clarify. Fig.1: This figure seems a

bit puzzling. The yellow line in Fig.1 (b) was explained as colony area, but the area included not only colony zone, but also ecotone zone. Please consider modifying the figure and legend.

---

## Author Comment (AC1) · 21 Feb 2018

bg-2017-492 Expansion of great cormorant colony immediately increased isotopic enrichment in small mammals Linas Balčiauskas, Raminta SkipitytÄŮ, Marius Jasiulionis, Laima BalčiauskienÄŮ, and Vidmantas Remeikis Handling Associate Editor: Sébastien Fontaine, sebastien.fontaine@clermont.inra.fr Status: Discussion (BG Discussions)

Anonymous Referee #1 However, I have some concerns about the design and the novelty of this study.

Comment: The first goal of this study as mentioned in Page 3 L 26 seems too obvious.

[Figure]

It should be clear that the nutrient input increases as the number of great cormorants and the faecal deposit become greater in the new habitat. I suppose that the authors intended to mention that they aimed to show how rapidly nitrogen derived from the great cormorants were used by primary produces and the consumers at higher trophic levels.

Answer: We thank Anonymous Referee #1, and extend the Aim (P3, L24-25) with "immediacy". Now Aim is formulated as "The aim was to evaluate immediacy of the effects of the transfer of biogens from the aquatic to terrestrial ecosystem by an expanding great cormorant colony, i.e., how rapidly nitrogen derived from the great cormorants was used by primary produces and the consumers at higher trophic levels.". Such formulation corresponds to the Abstract as well.

Comment: Even if this were one of their goals, the presented data would seem not enough to achieve the goal. The data used for this purpose were the isotopic signatures of two species of small mammals at a single control site in 2014 and at the same site but inhabited by the great cormorants in 2015. I think that replicated study sites would be necessary to examine statistically the effects of colony expansion on the isotopic signatures of consumers.

Answer: Unfortunately, replication was not possible. The colony is unique in Lithuania, and expansion was also unique event. Moreover, territorial expansion was mainly into the former control area! Number of rodents, trapped inside the zone, is finite (see Table 1 in the text).

Comment: In addition, the statistical results (Table 2) showed that there was almost no significant difference in 15N of the two small mammals between before-expansion (2014) and after-expansion (2015). It seems at least to me that the small mammals were not dependent on the cormorant-derived N, which is not consistent with the conclusion of this study (e.g., P2 L15). Please explain from which datasets the conclusion was drew. With the development of the great cormorant colony in 2015, the isotopic signatures, mostly $\delta$15N, in dominant small mammal hair grew compared to 2014, though not all differences are significant (Table 2).

Answer: Significance of differences depends on the sample size. Sample size was limited by situation, and without possibilities to be increased. Thus, results of Table 2 are restricted and final. However, we cannot agree with two things in this comment: 1. cormorant-derived N is well known as biogen in the colony, affecting all trophic levels (Ishida, 1996; Kameda et al., 2006; Klimaszyk and Rzymski, 2016; Nakamura et al., 2010) – references from this paper only. 20-30 references could be easily added to show nitrogen drastic increase in the colonies. 2. In Apodemus flavicollis (Table 2) $\delta$15N increased in ALL zones (increase in the ecotone zone, 7.5% is significant). Increase in the colony is ∼1%, expansion zone compared to former control – 5.7%, but all are correlated with colony growth and expansion. 3. in Myodes glareolus $\delta$15N increased in the ecotone zone, 2.3%, and in the colony, ∼1%. Not significantly, but still this is increase, and no other factor, just colony growth, could be expected. 4. yes, there was no increase of $\delta$15N in Myodes glareolus in the expansion zone. Please have in mind, that in both years single individuals were processed, as no more of them were present. We may expand text in P6 L27-28 if required.

Comment: Regarding the second goal, the authors had already shown that the great cormorant colonies significantly affected the isotopic signatures of the small mammals in their previous study published in 2016. Therefore, although I recognize the importance of datasets of the basal food sources presented by this study, the goal and the obtained results seem highly confirmatory.

Answer: For our best knowledge, this investigation is the first one to show speed/immediacy of the impact. To show immediacy, we need to compare results, even if some of them are already known. If Editors will advice, more results may be placed as Supplement.

Minor comments: - P2 L2: "and" damage Answer: agree, will be changed

- P2 L12: It was not surprising to me. It simply indicates that the plant used more cormorant-derived N than the invertebrates. Answer: Word "surprisingly" will be removed.

P3L14: It would be better to describe the background and the importance of this study based not on the authors' group interest, but on the scientific interest. Answer: fully agree with the comment. Proposed change to the text is: instead of "Our continuing interest in the great cormorant colony in JuodkrantÄŮ was maintained by the fact that it's area and number of breeding pairs increased in 2015 due to the absence of deterrent measures" will be changed to "However, area of the great cormorant colony in JuodkrantÄŮ and number of breeding pairs increased in 2015 due to the absence of deterrent measures"

P3L23: This sentence (gender and age) is not necessary here. Answer: will be removed

P5L2: If the authors think this information (Table 1) is necessary, please provide it as a supplementary file. Answer: If Editor is of the same opinion, we agree to remove.

P8L3: Which datasets showed the influences of great cormorants on small mammals? Please explain. Answer: there was no dataset presented, explanation in the text, P8 L6–10. Table with numbers of trapped individuals in 2014 and 2015 in different zones may be added as Supplement.

P8L24-P9L13: This subsection seems not directly related to the aims of this study. Answer: subsection was formulated to show, that various authors relate differences in stable isotope concentration to the diet. It arise after previous comments, to show that small mammal migrations had very limited influence, and to confirm, that isotopic differences reflect diet differences in the spatial zones of the cormorant colony.

P9L20: This sentence is unclear. Please clarify. Answer: Proposed change to the text is: instead of "So we support the opinion that cormorant influence is mediated

through disturbance of food resources (Millus and Stapp, 2008)." will be changed to "So we support the opinion of Millus and Stapp, 2008, that cormorant influence to small mammals is not direct, but is mediated through influence onto their food resources."

Fig.1: This figure seems a bit puzzling. The yellow line in Fig.1 (b) was explained as colony area, but the area included not only colony zone, but also ecotone zone. Please consider modifying the figure and legend. Answer: As we understand, most puzzling was legend, where "colony" was used for specific zone in the colony, and as "territory". We changed legend, and also changed picture, to show exactly, to where the colony was expanded, and how ecotone zone was in the area. In the territory, two areas inside were not inhabited by cormorants (we made colour more intense, to show this), thus, ecotone was in the border of one of such areas. Proposed change to the text: P4L13 instead of "Zone of ecotone was situated between colony and forest to the south (Fig 1a)." we propose "Zone of ecotone (Fig 1a) was situated between colony and unused forest inside the inhabited territory, shown in darker green in Fig 1b."

Also: Figure 1. Trapping design in the great cormorant inhabited territory in Juodkrant-tÄÅ̊ (a) and colony expansion in 2015 (b). Trapping of small mammals performed in the expansion zone (1), ecotone (2) and the colony (3). Yellow line – great cormorant inhabited territory in 2014, red line – colony expansion in 2015.

Please also note the supplement to this comment:
https://www.biogeosciences-discuss.net/bg-2017-492/bg-2017-492-AC1-supplement.pdf

---

## Referee Comment (RC2) · Anonymous Referee #2 · 15 May 2018

General comments:

First of all I have to precise that I am not familiar with the topic of biogenic pollution but with the use of 13C and 15N isotopic methods for other fields in ecosystems. However, such as they are presented in the introduction, the aims of this study sound very close to those of the previous paper published by the same authors in 2016. Perhaps consistent with this comment, the sentence ending the introduction (L28) was probably necessary to really indicate the novelty of this paper and then, its original aim: evaluating the speed of impact of great cormorant colony on small mammals. The results presented here partially confirm/reinforce the first results published in 2016 on

the impact of colony. However, I am not convinced by the methods (and a fortiori by the results) used for studying the speed of this impact, yet consisting in the main novel objective of this paper. This study emerged from a particular event where cormorant colony drastically and rapidly grew (2015) following several years of measures of limiting breeding success. I understand this consisted in an opportunity to test if colony expansion has rapid effect in this site. However it cannot help to quantify the speed of the effect but can only state if the effect can be rapid (1 year) or not, in this site. The design used does not allow statistical calculations (multiple sites) to generalize the effect of colony growth on isotopic signatures of small mammals. Moreover, Fig1 shows that the 3 zones (expansion, ecotone and colony) are partially confounded e.g. the ecotone zone is included in the colony zone - this point was unclear and very disturbing for me. Therefore, the scope of this study is strongly limited, not only regarding 1) the characterization of the speed of the effect (the main message) but also 2) its reliability to generalize the speed of colony effects in other sites. For these major reasons I mainly perceived this manuscript as a complement of the former paper (2016) rather than a novel paper addressing a research on the speed of the cormorant colony effects.

Detailed comments: Introduction should better develop scientific implications and questions emerging from studying colony impact. What are the consequences in terms of scientific interests? P3L18-24. This half-paragraph is focused on the approach. It should be shortened here and be developed more extensively in the material & methods section. P4L13: Replace It's by Its P8L18: "Stable" is repeated twice, remove one P9L1-2: This sentence sounds redundant with previous paragraph. Maybe it could be included in previous paragraph. Table2: horizontal alignment should be modified

---

## Author Comment (AC2) · 21 May 2018

Dear Editor, dear anonymous Reviewer#2, please find answers to comments. In the name of all authors Linas Balčiauskas

Biogeosciences Discuss., https://doi.org/10.5194/bg-2017-492-RC2, 2018 © Author(s) 2018. This work is distributed under the Creative Commons Attribution 4.0 License. Answers to Interactive comment on "Expansion of great cormorant colony immediately increased isotopic enrichment in small mammals" by Linas Balčiauskas et al. Anonymous Referee #2

General comments: Comment: First of all I have to precise that I am not familiar with the topic of biogenic pollution but with the use of 13C and 15N isotopic methods for other fields in ecosystems. However, such as they are presented in the introduction, the aims of this study sound very close to those of the previous paper published by the same authors in 2016. Perhaps consistent with this comment, the sentence ending the introduction (L28) was probably necessary to really indicate the novelty of this paper and then, its original aim: evaluating the speed of impact of great cormorant colony on small mammals. The results presented here partially confirm/reinforce the first results published in 2016 on the impact of colony. However, I am not convinced by the methods (and a fortiori by the results) used for studying the speed of this impact, yet consisting in the main novel objective of this paper. This study emerged from a particular event where cormorant colony drastically and rapidly grew (2015) following several years of measures of limiting breeding success. I understand this consisted in an opportunity to test if colony expansion has rapid effect in this site. However it cannot help to quantify the speed of the effect but can only state if the effect can be rapid (1 year) or not, in this site.

Answer: this comment has much in common with the comment of Rev#1, thus, we dare to answer in short way for not to repeat it. Novelty of the presented manuscript is in evaluation of the immediacy of cormorant influence to mammals – such results, to our best knowledge, are presented for the first time. In fact, there is even not much published information on the small mammal ecology changes under influence of the Great Cormorant colony. Thus, even agreeing with the fact, that presented results re-inforce already published tendencies (Balciauskas et al., 2016), we see sufficient input to the science and practice in the presented manuscript. Scientifically, it is first time that immediacy of the cormorant colony is confirmed, in practice these results show, that cormorant scarring may yield not expected results. Namely, terrestrial ecosystem in influenced in the new places as cormorants move their nests, and influence is im-mediate, in the same year. We suppose to add "in the same year" to the manuscript text. We also are going to add text to P10L26, explaining unwanted practical effect of

the cormorant scarring from the colony.

Comment: The design used does not allow statistical calculations (multiple sites) to generalize the effect of colony growth on isotopic signatures of small mammals. Moreover, Fig1 shows that the 3 zones (expansion, ecotone and colony) are partially confounded e.g. the ecotone zone is included in the colony zone - this point was unclear and very disturbing for me.

Answer: we fully agree with comment, that investigation was restricted to single site. Unfortunately, replication was not possible. The colony in JuodkrantÄŮ is unique in Lithuania, and its expansion was also unique event, never happened before. Moreover, territorial expansion was mainly into the former control area! Thus, we fortunately had data from the former control zone. However, number of rodents, trapped inside the zone, is finite (see Table 1 in the text). Trapping cannot be extended, as size of the colony is limited, density of the rodents, as we already stated in previous publications, is limited, and only two species have numbers, giving an opportunity to test differences between colony zones. We had no chances to choose expansion zone, as cormorants themselves settled in the previous control zone after scaring measures ceased. As for statistics, to show immediacy, we need to compare results from previous year, even if some of them are already known. Sample size could not be increased due to completely objective reason (size of the colony and limited number of rodents), thus, we did our best. Still we found, that in Apodemus flavicollis (Table 2) $\delta$15N increased in ALL zones (increase in the ecotone zone, 7.5% is significant). Increase in the colony is ∼1%, expansion zone compared to former control – 5.7%, but all values are correlated with colony growth and expansion. It is easy to calculate, that ca. 40% bigger sample would have significant differences in all comparisons. As there are no other suspected factors, just number of birds (nest appearance) and their biological pollution, we found such increase worth to analyse. However, we have no data from other published sources for comparison – no such publications were found. Fig. 1 was already reworked, as shown in the answer to Rev#1. Ecotone is not in the colony itself

– it is between colony and unused forest, just unused forest is irregularly shaped and partially surrounded by colony. We expect that after changes it is clear to the reader.

Comment: Therefore, the scope of this study is strongly limited, not only regarding 1) the characterization of the speed of the effect (the main message) but also 2) its reliability to generalize the speed of colony effects in other sites. For these major reasons I mainly perceived this manuscript as a complement of the former paper (2016) rather than a novel paper addressing a research on the speed of the cormorant colony effects.

Answer: we agree that this study is limited in many aspects, including lack of repeated measurements; however, it is novel in the aspect of assessing immediacy of the influence of great cormorant colony to small mammals! We cannot agree with Reviewer#2 opinion about possible generalization, as there are no chances to test both (his and our) opinions. We may just hypothesize that publication of these results urge scientists of testing immediacy in other sites, if growth of the colony (including other colonial birds) will be available.

Comment: Introduction should better develop scientific implications and questions emerging from studying colony impact. What are the consequences in terms of scientific interests?

Answer: In the introduction we focused on small mammals, living in the Great cormorant colony. P3L10-14 show, that most of the biology of small mammals is affected. However, so far we investigated consequences of the long-term impact. We expect that showing immediacy of the impact of the colonial bird gives new insight into science of ecosystem change, explaining, that in extreme cases rapid changes may occur due to natural causes, such as increase of biological pollution. From the practical implication, results let us conclude, that management of the colonies of great cormorants may have unexpected outcome: if scared birds start to nest in the new areas, they affect it up to mammal level in the first year. In the protected areas with valuable habitats around

the colony such situation may be completely unacceptable. Instead, inhibiting of the growth of bird numbers by limiting breeding success may be preferred.

Comment: P3L18-24. This half-paragraph is focused on the approach. It should be shortened here and be developed more extensively in the material & methods section.

Answer: we follow suggestion of rev#2. Former text "Measures of limiting breeding success in JuodkrantÄŮ great cormorant colony were started in 2004 (Knyva, unpublished) and they withhold colony from expansion. In 2015 measures were not applied, resulting colony growth. First nests appeared in the area, which was free of cormorants in 2014, thus, was used as control zone in Balčiauskas et al., (2016). In 2015 we repeatedly examined $\delta$13C and $\delta$15N distribution in a great cormorant colony, this time including in plants and invertebrates as expected diet sources of small mammals." was removed from Introduction, shortened, and incorporated after P4L10. After changes, text in P4L4-9 is: "In 2004 number of breeding pairs reached 2800. In the same year measures of limiting breeding success in JuodkrantÄŮ great cormorant colony were started (Knyva, unpublished) and they withhold colony from expansion. Over 3500 nests have been recorded in the colony each year since 2010, with the exception of 2014 when, due to control measures (firing petards in the nesting period), the number of successful pairs was under 2000. In 2015 measures were not applied, resulting colony growth. First nests appeared in the area, which was free of cormorants in 2014, thus, was used as control zone in Balčiauskas et al., (2016)."

P4L13: Replace It's by Its Answer: corrected

P8L18: "Stable" is repeated twice, remove one Answer: corrected

P9L1-2: This sentence sounds redundant with previous paragraph. Maybe it could be included in previous paragraph. Answer: we follow suggestion of Rev#2, moving sentence to previous paragraph

Table2: horizontal alignment should be modified Answer: cells re-aligned top left

Please also note the supplement to this comment:
https://www.biogeosciences-discuss.net/bg-2017-492/bg-2017-492-AC2-
supplement.pdf

---

## Author Response (AR1)

Answer to Associate Editor Decision: Reconsider after major revisions (23 May 2018) by Sébastien Fontaine

Comments to the Author: The comments of the two referees made a number of interesting comments with the aim of improving your work. Their comments are united in some aspects such as the lack of clarity on the novelty of this article compared to your previous publication and the lack of robustness of your statistical approach.

You carefully answered to these comments. In one place you suggest that we could add some new data to reinforce your main message. Since the two referees question the robustness of your approach I invite you to add all the data that could reinforce your work when you will submit your revised version.

**Dear Sébastien,**

We present addition to the text on the increase of stable isotope values, comparing zone of the great cormorant colony in 2014 and 2015 (where former control zone became expansion zone). As we wrote in the answer to reviewers, sample size cannot be increased due to objective reasons, so there is limited possibility for statistics. However, we see no other factor, just colony growth and spread, involved in the observed changes. Additionally we checked cores areas of species in the isotopic space – if they remain the same under increase of number of breeding cormorants and after they appear in new area.

**Added text: P7L9-18**

"With the expansion of the great cormorant colony in 2015, the isotopic signatures of  $\delta^{15}$ N in dominant small mammal hair grew in comparison to 2014, though not all differences are significant (Table 2). In *A. flavicollis*,  $\delta^{15}$ N values increased in all zones (the 7.5 % increase in the ecotone zone is significant at p < 0.05). The increase in the colony is ~1 %, while the expansion zone compared to former control zone is 5.7 %. All are correlated with colony growth and expansion. In *M. glareolus*,  $\delta^{15}$ N increased by 2.3 % in the ecotone zone and ~1 % in the colony.

 $\delta^{13}$ C signatures in the hair of *A. flavicollis* in 2015 decreased in all zones. The decrease in the expansion zone compared to 2014 control zone was 4.5 % (p < 0.1), in the colony zone 3.1 % (p < 0.001) and in the ecotone zone 0.5 %. In the hair of *M. glareolus*, the decrease of  $\delta^{13}$ C signatures was even stronger - 8.5 % in the expansion zone, 3.3 % (p < 0.1) in the control zone and 2.6 % in the ecotone (Table 2). We suppose that no other factorother than colony growth could account for these changes."

To evaluate significance of these changes, we additionally tested data on stable isotopes, calculating ellipses of species (core areas in the isotopic space) using SIBER package run in R (according Jackson et al., 2011). Non-intersecting ellipses would show, that position in the isotopic space is significantly shifted (thus, confirming significance of the influence of colony impact in 2015 compared to 2014). Package is using Bayesian method, but even so for the graphic representation of ellipses and core areas it requires sample size n=5. This was not fulfilled for *Myodes glareolus* in the control/expansion zone, where 1-2

individuals were trapped (see added Table S1). In this case central position (mean±SE, ‰) of stable isotope ratios in the hair of *Myodes glareolus* could be seen in the Table 2.

**Added text: P7L18-22**

"Isotopic niches of *A. flavicollis* in 2014 and 2015 (shown as central ellipses) were separated in the colony and between the control and expansion zones, while they partially overlapped in the ecotone (Fig. S2a). The isotopic niches of *M. glareolus* in 2014 and 2015 were separated in the ecotone and had a small overlap in the colony (Fig. S2b). Insufficient sample size did not allow analysis in the control and expansion zones for this species (see Table 1)."

We add four supplements. Fig. S1 is showing normality of the distribution of stable isotopes (two small mammal species,  $\delta^{15}$ N and  $\delta^{13}$ C, zones lumped). Figure shows, that namely expansion zone had outliers from normal distribution of  $\delta^{13}$ C, while  $\delta^{15}$ N distribution is not normal.

In the Fig. S2 – changes of core areas in the isotopic space of both small mammal species are shown according zones of the great cormorant colony, comparing 2014 and 2015.

Fig. S3 shows central position of stable isotope ratios in litter, cormorant feathers and the hair of *Apodemus flavicollis* and *Myodes glareolus* in different zones of the Juodkrantė colony of great cormorants in 2015.

Introduced Table1 now show number of trapped individuals of both species according zones of the colony in 2014 and 2015. Due to various reasons, not all trapped small mammals were analysed for isotope concentrations in 2014. In 2015, we took hair from nearly all individuals, excluding only those already damaged by insects. To show, that sample size is limited by objective reason, we decided to keep Table 1 in the text. For the Rev#2 such table did not raise objections or comments. New Table 1 also answer comment of Rev#1 for P8L3 "which datasets shoved the influences of great cormorants on small mammals?" Former Table 1 now is presented as supplement, thus we answer comment of Rev#1 to give former Table 1 as Supplement. And so, 2.2 subchapter was re-composed.

To put more emphasis on the immediacy of the observed changes, we re-phrased title of the paper, now it is "Immediate increase in isotopic enrichment in small mammals following the expansion of a great cormorant colony."

Language was re-checked, adjustments done by native speaker. We also added Acknowledgements.

We hope these changes not only answer all remaining questions, but also improve quality of the manuscript.

With best wishes

In the name of all authors

Linas Balčiauskas

**Expansion of great cormorant colony immediately increased Immediate increase in isotopic enrichment in small mammals following the expansion of a great cormorant colony**

Linas Balčiauskas1, Raminta Skipitytė1,2, Marius Jasiulionis1, Laima Balčiauskienė1, Vidmantas Remeikis2

[revised manuscript text omitted]
 lone year; with most over-winter mainlywintering individuals being autumn-born ones (Bobek, 1969), thus, our
samples thus represent cormorant the influence of the cormorants in the year when that the rodents were trapped.

- 10 samples thus represent cormorantthe influence of the cormorants in the year whenthat the rodents were trapped. Environmental samples (including plants, litter, invertebrates and great cormorant feathers and eggs) were stored in a refrigerator at below -20°C prior to preparation and analysis. Samples were dried in an oven at 60°C to a constant weight for 24–48 hours and then homogenized to a fine powder (using mortar and pestle and a Retsch mixer mill MM 400). Pretreatment of hair and other samples was not useddone, as after testing it gaveproduced no change of results. Feathers were cleaned with
- 15 acetone and deionized water prior to measurements. Feather samples were clipped from the vane avoiding the rachis. Stable isotope ratios ( $\delta^{13}$ C and  $\delta^{15}$ N) were measured using an elemental analyzer (EA) coupled to an IRMS (Flash EA1112; Thermo Delta V Advantage, Thermo Scientific, USA). Stable isotope data are reported as  $\delta$  values, according to the formula  $\delta X = [R_{sample}/R_{standard} - 1] \times 10^3$ , where  $R_{sample} = {}^{13}C/{}^{12}C$  or  ${}^{15}N/{}^{14}N$  of the sample,  $R_{standard} = {}^{13}C/{}^{12}C$  or  ${}^{15}N/{}^{14}N$  of the standard. 5 % of samples were run in duplicate. The equipment parameters and measurement quality control are detailed elsewhere
- 20 (Balčiauskas et al., 2016).

**2.5 Statistical analysis**

Normality The normality of distribution of  $\delta^{13}$ C and  $\delta^{15}$ N values was tested using Kolmogorov-Smirnov's D. As not all values of  $\delta^{13}$ C and  $\delta^{15}$ N were distributed normally, the influences of species and the zone of the colony on the carbon and nitrogen stable isotope values in the mammal hair were tested using nonparametric Kruskal-Wallis ANOVA. Independent groups were

- compared with the same Kruskal-Wallis multiple comparisons procedure (Electronic, 2017). Differences in  $\delta^{13}$ C and  $\delta^{15}$ N ratios between 2014 (data from Balčiauskas et al., 2016) and 2015 were tested by multivariate Hotteling's T2 test. The minimum significance level was set at p < 0.05. Calculations were performed using Statistica for Windows, ver. 6.0. Environmental samples were analyzed by object group (cormorant, litter, invertebrates, plants) and by the zone (expansion, ecotone, colony). Isotopic baselines were calculated using basal resources as possible foods for rodents grouped according to
- 30 their origin. Reported values are arithmetic means with SE of the  $\delta^{15}N$  and  $\delta^{13}C$  for all basal resources mentioned above.

The isotopic niches of species, as central ellipses, were calculated using SIBER (Jackson, Inger, Parnell, & Bearhop, 2011) under R ver. 3.5.0 (https://cran.r-project.org/bin/windows/base/rdevel.html) for *A. flavicollis* and *M. glareolus* in the zones, where five or more individuals were investigated for  $\delta^{15}$ N and  $\delta^{13}$ C.

**3 Results**

**5 3.1 $\delta^{13}$ C and $\delta^{15}$ N values in the hair of small mammals inhabiting the great cormorant colony**

Distribution The distribution of both  $\delta^{13}$ C and  $\delta^{15}$ N values in the hair of *A. flavicollis* was not normal, while in *M. glareolus*-the distribution of  $\delta^{15}$ N values in *M. glareolus* was also not normal, but the distribution  $\delta^{13}$ C values corresponded to normal (Fig. S1). Outliers from the normal distribution were values, registered in the expansion zone. (Fig. S1). Kruskal-Wallis ANOVA demonstrated that the distribution of stable isotope values was influenced not only by zone, but also by the species of small

10 demonstrated that the distribution of stable isotope values was influenced not only by zone, but also by the species of small mammal. These factors together significantly influenced the distribution of  $\delta^{15}N$  (r2 = 0.31) and  $\delta^{13}C$  (r2 = 0.26, F both p < 0.0001).

In 2015, the influence of the zone (both species pooled) was significant for the distribution of  $\delta^{15}$ N (Kruskal-Wallis ANOVA, H2.119 = 18.62, p = 0.0001) and  $\delta^{13}$ C (H2.119 = 6.30, p = 0.043). Between-species differences of the stable isotope values in

15 the hair of the rodents in the colony area were highly significant for  $\delta^{13}$ C (H1,119 = 21.69, p < 0.0001) and for  $\delta^{15}$ N (H1,119 = 6.67, p = 0.01).  $\delta^{15}$ N values were highest in the hair of *M. glareolus* trapped in the ecotone and colony zones, while highest in *A. flavicollis* in the expansion zone.  $\delta^{13}$ C signatures in the hair of *A. flavicollis* were higher than in *M. glareolus* in all territories, including the expansion zone (Table 2).

With the development expansion of the great cormorant colony in 2015, the isotopic signatures, mostly of  $\delta^{15}N_7$  in dominant

- small mammal hair grew compared in comparison to 2014, though not all differences are significant (Table 2). In *A. flavicollis*,  $\delta^{15}$ N values increased in all zones (the 7.5 % increase in the ecotone zone is significant at p < 0.05). The increase in the colony is ~1 %, while the expansion zone compared to former control zone is 5.7 %. All are correlated with colony growth and expansion. In *M. glareolus*,  $\delta^{15}$ N increased by 2.3 % in the ecotone zone and ~1 % in the colony.  $\delta^{13}$ C signatures in the hair of *A. flavicollis* in 2015 decreased in all zones. The decrease in the expansion zone compared to
- 25 2014 control zone was 4.5 % (p < 0.1), in the colony zone 3.1 % (p < 0.001) and in the ecotone zone 0.5 %. In the hair of *M*. *glareolus*, the decrease of δ13C signatures was even stronger 8.5 % in the expansion zone, 3.3 % (p < 0.1) in the control zone and 2.6 % in the ecotone (Table 2). We suppose that no other factorother than colony growth could account for these changes. Isotopic niches of *A. flavicollis* in 2014 and 2015 (shown as central ellipses) were separated in the colony and between the control and expansion zones, while they partially overlapped in the ecotone (Fig. S2a). The isotopic niches of *M. glareolus* in
   20 2014 and 2015 are separated by the second decound be and the second decound decound be and the second decound decound decound be and the second decound de

[revised manuscript text omitted]